# Characterization of the Sensory Properties and Quality Components of Huangjin Green Tea Based on Molecular Sensory-Omics

**DOI:** 10.3390/foods12173234

**Published:** 2023-08-28

**Authors:** Ni Zhong, Xi Zhao, Penghui Yu, Hao Huang, Xiaocun Bao, Jin Li, Hongfa Zheng, Lizheng Xiao

**Affiliations:** 1Key Laboratory of Tea Science of Ministry of Education, Hunan Agricultural University, Changsha 410128, China; daren_ni@163.com (N.Z.);; 2National Research Center of Engineering Technology for Utilization of Functional Ingredients from Botanicals, Hunan Agricultural University, Changsha 410128, China; 3Tea Research Institute, Hunan Academy of Agricultural Sciences, Changsha 410128, China

**Keywords:** Huangjin green tea, sensory characteristic, liquor color, aroma, taste

## Abstract

Huangjin green tea (HJC) is one of the most famous regional green teas in China, and has gained attention for its unique flavor. Research on HJC has focused mainly on the synthesis of L-theanine, with fewer studies concentrating on sensory characteristics. In this study, molecular sensory science techniques, including color analysis, gas chromatography–ion mobility spectrometry, and E-tongue, were used to characterize the sensory properties of HJC, with Fuding Dabai and Anji Baicha teas used as conventional and high amino acid controls, respectively. The sensory characteristics and main quality components of HJC lie somewhere between these two other teas, and somewhat closer to the conventional control. They were difficult to distinguish by color, but significant differences exist in terms of volatile organic compounds (VOCs), E-tongue values on bitterness and astringency, and their contents of major taste components. VOCs such as (E)-2-octenal, linalool, ethyl acrylate, ethyl acetate, and 2-methyl-3-furanethiol were found to be the main differential components that contributed to aroma, significantly influencing the tender chestnut aroma of HJC. Free amino acids, tea polyphenols, and ester catechins were the main differential components responsible for taste, and its harmonious phenol-to-ammonia ratio was found to affect the fresh, mellow, heavy, and brisk taste of HJC.

## 1. Introduction

Green tea is an unfermented tea that is widely produced in East Asian countries and is gaining popularity worldwide, mainly for the quality of its flavor [1,2] and its potential health benefits [3,4,5]. The comprehensive quality of tea infusions is mainly reflected in color, aroma, and taste [6]. The bright yellow-green color of the infusions, along with their umami-rich, brisk taste and fragrant aroma, are indicative of the all-encompassing quality of green tea. It has been discovered that the aroma of green tea varies between different products, and consumers can be attracted to teas with certain charming and/or pleasing aromas. For instance, a baked-like, chestnut-like aroma is found in Xihu Longjing, an orchid-like aroma is present in Taiping Houkui, and a corn-like aroma is prominent in Huangjinya [7,8,9]. Extensively researched in various green teas, key differential volatile organic compounds (VOCs) have been investigated. For example, VOC compounds such as (Z)-jasmone, linalool, geraniol, β-cyclocitral, β-ionone, (E)-lactone, and indole have been defined as signature green tea odorants [10,11,12,13]. Taste, which depends primarily on the chemical composition and content of tea products, is another essential indicator that determines the quality of tea, as well as an important driver of consumer acceptance. To date, it has been established that polyphenols, amino acids, and caffeine are the primary contributors to the taste. The bitterness and astringency of tea infusions are largely due to the presence of tea polyphenols and caffeine. Catechins, which make up 70–80% of total tea polyphenols, are usually responsible for these flavors [14]. Amino acids, however, are a major factor in determining the umami of green tea and may contribute to reducing the astringency and bitterness caused by polyphenols and caffeine [15].

Different teas from around the world exhibit different characteristics that stem from different environments, varieties, cultivations, seasons, and manufacturing processes [16,17,18,19]. Due to the relatively simple processing that green tea undergoes, which includes steps such as fixation, rolling, and drying, it tends to retain more natural metabolites than other types of tea products [20]. As a result, studies have revealed that variety is the primary factor influencing the qualities of green tea, with clear infusions, green leaves and a strong flavor convergence [21,22]. It has been reported that Xihu Longjing tea, processed from ‘Longjing 43’, best matches the characteristics of its product’s color, due to its high chlorophyll b content, which is conducive to the formation of a brown-beige color [23]. Xinyang Maojian tea, made from ‘Xinyang Quntizhong’ that grows in the northernmost area of China, has a unique flavor that is related to its key odorants. These include volatile terpenes and fatty acid-derived volatiles, which decrease significantly following fixation [24]. ‘Shidacha’ has been reported to be the most suitable variety for making Taiping Houkui tea, mainly due to the formation of the methyl jasmonate phenotype in its fresh leaves during processing, which contributes to its strong orchid fragrance [25]. Enshi green tea, made from ‘Echa 10’, imparts a special honeysuckle fragrance, which has been found to be attributable to key components such as dodecane, octadecane, phenethyl alcohol, and jasmonone [19]. Anji baicha, made from ‘Baiye 1’, a representative albino variety, has a fresh and mellow taste that derives from its high content of total free amino acids and low polyphenols [26]. ‘Fuding Dabai’ has been recognized as a national improved variety by the China Tea Variety Approval Committee. Its versatility for planting in a variety of regions and its capacity for processing are both highly desirable. The green tea processed from its leaves is usually defined as a standard reference [27].

The main production area for Huangjin green tea (HJC) is in the Xiangxi Autonomous Prefecture of the Hunan Province, China. The tea industry has become a major source of agricultural income in this region, spanning an area of about 40,000 hectares. Due to its outstanding quality, the tea has become the best-selling tea variety in Hunan. The resource used to make HJC, ‘Huangjincha’, is a rare and ancient local tea germplasm in China that originated in Baojing County of the Xiangxi Autonomous Prefecture. It has four major industrial competitive advantages, including early germination, high yield, excellent quality, and strong suitability. As such, it is favored by many tea farmers and consumers alike. At present, high-quality ‘Huangjincha’ resources are constantly being selected and registered as national excellent varieties, among which the ‘Baojing Huangjincha 1’ variety was selected as China’s leading agricultural variety in 2023, due to its outstanding characteristics. At this time, research surrounding Huangjincha has mainly focused on the synthesis of L-theanine in tea plants, as a high amino acid tea variety [28,29]. Far fewer studies have focused on comprehensive evaluations of its high-quality sensory characteristics.

The Standardization Administration of China has issued the “GB/T 23776–2018 Methodology for sensory evaluation of tea” as the most prevalent and essential assessment technique for green tea. Nevertheless, there are certain clear drawbacks to this method. The results are susceptible to individual subjectivity and environmental interference. The advantages of high sensitivity, accuracy, and speed of modern instruments, which are not influenced by personal subjective factors are such that it is necessary to establish a digital sensory evaluation system by combining traditional sensory review with analysis of physical and chemical sensory indices through a variety of intelligent sensory technologies. 

Therefore, the aim of this study was to comprehensively characterize the sensory characteristics of early spring HJC made from fresh leaves of the ‘Baojing Huangjincha 1’ cultivar, using molecular sensory-omics techniques such as color difference analysis, GC–IMS, and electronic tongue, and to reveal the differences in color, aroma, and taste between HJC and the control samples, namely, conventional green tea Fuding Dabai (FD) and high amino acid green tea Anji Baicha (AJ). The results obtained from this study will provide a scientific explanation for the sensory characteristics of green tea as well as a reference for normalizing, standardizing and for the scientization of the sensory assessment of tea products.

## 2. Materials and Methods

### 2.1. Sample Collection

Twenty-seven green tea samples with typical characteristics were directly selected from local tea factories in the Hunan and Zhejiang provinces of China, in March of 2021, by experienced tasters. These comprised: an HJC obtained from the Xiangxi Autonomous Prefecture, Hunan Province (samples HJC1–HJC9), an FD obtained from Hunan Province outside the Xiangxi Autonomous Prefecture (FD1–FD9), and an AJ obtained from Anji County in the Zhejiang Province. Appendix A furnishes an in-depth look at the tea samples. All of them were securely sealed and kept in a refrigerator at a temperature of −20 °C for further examination.

### 2.2. Chemicals and Reagents

Samples of (+)-catechin (DL-C), (−)-epicatechin (EC), (−)-gallocatechin (GC), (−)-epigallocatechin (EGC), (−)-epicatechin gallate (ECG), (−)-epigallocatechin gallate (EGCG), (−)-gallocatechin gallate (GCG), and caffeine were acquired from Mansit Biotechnology (Chengdu, China) in pure form. Merck (Darmstadt, Germany) supplied glutamic acid, methanol, and N, N-Dimethylformamide (DMF). All chemicals used for liquid chromatography in this study were of chromatographic grade. Analytically pure chemical reagents were procured from MACKLIN Corporation in Shanghai, China.

### 2.3. Traditional Sensory Evaluation

Seven professional tea tasters (three females and four males, aged 30–60 years) from the Hunan Agriculture University or Tea Research Institute (Hunan, China) were consulted to assess and evaluate sensory characteristics. All panelists had >10 years of descriptive sensory analysis experience with tea. In accordance with the Chinese national standard procedure for evaluating tea leaves (GB/T 23776–2018), 150 mL of boiling water was added to 3 g of each tea sample in separate teacups with their respective lids shut for 4 min, to obtain the tea infusions. The panelists then evaluated the appearance (25%), liquor color (10%), aroma (25%), taste (30%), and the infused leaves (10%) of each tea, gave comments, and scored the samples. Finally, the total scores of their organoleptic qualities were calculated based on their weight values. Parallel experiments were performed in triplicate.

### 2.4. Liquor Color Determination

The tea infusions were prepared according to the sensory evaluation method. Quantifying tea infusions with the International Commission on Illumination (L*, a*, b*) system, L* symbolizes lightness, a* stands for red (+a*) and green (−a*), and b* stands for yellow (+b*) and blue (−b*) [30]. Each measurement was performed in triplicate. A portable colorimeter (SMY–2000, Shengmingyang Science and Technology Development Co., Ltd., Beijing, China) was used to determine the color difference attributes that could be used to reflect the degree of color of each sample. Equations (1)–(4) are formulated by the following formulae: the variables ΔL*, Δa*, and Δb* denote chromatic aberrations, while the ΔE* value stands for the total chromatic aberration.
ΔL* = L*_M_ − L*_0_;(1)
Δa* = a*_M_ − a*_0_;(2)
Δb* = b*_M_ − b*_0_;(3)
ΔE* = (ΔL*^2^ + Δa*^2^ + Δb*^2^)^1/2^,(4)

In the above Equations, L*_M_ is the L* value of the sample, and L*_0_ is the L* value of the standard sample (HJC1); a*_M_ is the a* value of the sample, and a*_0_ is the a* value of the standard sample (HJC1); and b*_M_ is the b* value of the sample, while b*_0_ is the b* value of the standard sample (HJC1).

### 2.5. Volatile Organic Compounds

GC–IMS, an Agilent 490 gas chromatograph (Agilent Technologies, Palo Alto, CA, USA) and an IMS instrument (FlavourSpec^®^, Gesellschaft für analytische Sensorsysteme GmbH, Dortmund, Germany) were employed to ascertain VOCs in three varieties of early spring green tea samples. The relevant parameters were set according to those used in an earlier study by Xu et al. [31], with slight modifications. Samples of 1 g tea, weighed precisely and sealed in a 20 mL headspace glass, were then incubated at 60 °C for 10 min. Subsequently, 500 μL of headspace volume was injected into the instrument at 85 °C. The compounds were then separated by a capillary column (FS–SE–54–CB–1, 15 m × 0.53 mm, 1.0 μm) at a temperature of 60 °C. At 45 °C, the draft tube was utilized with nitrogen as the drift gas (purity ≥ 99.999%) at a rate of 150 mL/min. The programmed inlet gas was then circulated in the following order: 0–2 min, 2 mL/min; 2–10 min, 10 mL/min; 10–20 min, 100 mL/min; and 20–30 min, 150 mL/min. The volatile compounds were identified based on drift times and retention indices, using standards from the NIST and G.A.S. databases. All measurements were done in triplicate.

### 2.6. Taste Evaluation

The E-tongue system (TS–5000Z, INSENT, Atsugi, Japan) was equipped with six taste sensors, including AAE, CTO, CAO, COO, AE1, and GL1, as well as a reference electrode (R). The six basic tastes of umami, saltiness, sourness, bitterness, astringency, and sweetness, as well as astringent aftertaste (After–a), bitter aftertaste (After–b), and umami aftertaste (richness), are all measured by this instrument. The taste sensors are composed of a lipid-based membrane, an Ag/AgCl electrode, and an internal cavity filled with a 3.33 M KCl aqueous solution and saturated AgCl. As references, 30 mM KCl and 0.3 mM tartaric acid were tested. The sensory evaluation method was followed for each tea infusion, with the temperature of the room being cooled to around 25 °C for potentiometric measurements. This process included cleaning the positive and negative electrodes for a period of 90 s, two cleanings of the reference solution for 120 s each, sample detection for 30 s, and “after-taste” detection for another 30 s. Each tea sample was infused in triplicate on the same day, and each infusion was measured four times.

### 2.7. Determination of Tea Quality Components

Ground samples of each tea, of a weight of 3 g, were placed in a 500 mL triangular flask. Boiling water was added, and the sample was placed in a 100 °C water bath for 45 min, shaking it every 10 min. The tea broth was then filtered while still hot, and the filtrate was transferred to a 250 mL volumetric flask, cooled, and fixed before being measured.

Total soluble solids were measured in accordance with the State Standard of China GB/T 8305–2013 [32]. Total tea polyphenols (using the ferrous tartrate spectrophotometric method), total free amino acids (using the ninhydrin colorimetric method), and total soluble sugars (using the anthrone-sulfuric acid colorimetric method) were measured according to the procedure outlined in a previous report [33]. All measurements were done in triplicate. Specific steps are as follows.

The prepared tea infusion (1 mL), distilled water (4 mL), and reaction solution [0.1% FeSO_4_ and 0.5% potassium sodium tartrate (C_14_H_4_O_6_KNa·4H_2_O)] (5 mL) were put into a 25 mL volumetric flask in sequence, and the volume made up with phosphate buffer (1/15 M Na_2_HPO_4_, 1/15 M KH_2_PO_4_, pH 7.5). Absorbance values were read at 540 nm in a UV spectrophotometer. Total tea polyphenols content was calculated using Equation (5).
(5)$$\mathrm{Total}\;\mathrm{tea}\;\mathrm{polyphenols}(\%)=\frac{A\times 3.914}{1000}\times \frac{L1}{L2\times M\times m}\times 100$$

L1—the total volume of the tea infusion, mL;

L2—the volume of the infusion taken to reaction, mL;

M—the dry weight of the tea sample, g;

m—the dry ratio of the tea sample, %;

3.914—factor denoting that 1 A using the 10 mm color comparison cell was equal to 3.914 mg of the tea polyphenols in the tea infusion.

The prepared tea infusion (1 mL), phosphate buffer (1/15 M Na_2_HPO_4_, 1/15 MKH_2_PO_4_, pH 8.0) (0.5 mL), and reaction solution [2% ninhydrin (C_9_H_4_O_3_·H_2_O)] (0.5 mL) were put into a 25 mL volumetric flask in sequence, then boiled for 15 min. After cooling down, the volume was made up with distilled water. Absorbance values were read at 570 nm in a UV spectrophotometer. The total free amino acids content was calculated using Equation (6).
(6)$$\mathrm{Total}\;\mathrm{free}\;\mathrm{amino}\;\mathrm{acids}(\%)=\frac{\frac{\mathrm{CV}1}{1000\times V2}}{Mw}\times 100$$

C—the total free amino acids weight (mg), which could be obtained according to the OD_570_ from a standard curve made by theanine or glutamic acid as a standard component, using the same method as mentioned above;

V1—the total volume of the tea infusion, mL; 

V2—the volume of the infusion taken to reaction, mL; 

M—the dry weight of the tea sample, g; 

w—the dry ratio of the tea sample, %. 

The prepared tea infusion diluted 5 times (1 mL) and reaction solution [0.6 g anthrone (C_14_H_10_O), 100 mL sulfuric acid(H_2_SO_4_)] (8 mL) were put into a 25 mL volumetric flask in sequence. Boiled for 3 min, it was then immediately placed in an ice bath to cool to room temperature. Absorbance values were read at 620 nm in a UV spectrophotometer. The total soluble sugars content was calculated using Equation (7).
(7)$$\mathrm{Total}\;\mathrm{soluble}\;\mathrm{sugars}(\%)=\frac{\frac{\mathrm{AV}1}{1000}\times D}{V2\times \mathrm{Mw}}\times 100$$

A—the total soluble sugars weight (mg), which could be obtained according to the OD_620_ from a standard curve made by glucose as a standard component, using the same method as mentioned above;

V1—the total volume of the tea infusion, mL; 

V2—the volume of the infusion taken to reaction, mL; 

D—dilution ratio;

M—the dry weight of the tea sample, g; 

w—the dry ratio of the tea sample, %.

High-performance liquid chromatography (HPLC; Shimadzu, Tokyo, Japan) was employed to ascertain the catechins and caffeine contents. Prior to injection, a 0.45 μm filter was used to filter the infusions. The DiamonsiTM C18 column (4.6 mm × 250 mm, 5 μm; Dikma Technologies, Foothill Ranch, CA, USA) was employed, with the column temperature set at 37 °C. The mobile phases were composed of eluent A (water) and eluent B (DMF/methanol/acetic acid = 39.5/1.5/1 [V/V/V]). Beginning with 14% B, the linear gradient was conducted for 0–13 min, 23% B for 13–28 min, 36% B for 28–34 min, and finally 14% B for 34–43 min. The flow rate was 1 mL/min and the detection wavelength was 280 nm.

### 2.8. Data Statistics and Analysis

Significant differences in relative color values, E-tongue indicators, and relative concentrations of VOCs and main quality components were analyzed using SPSS v20.0. Principal component analysis (PCA) and partial least squares discriminant analysis (PLS–DA) were carried out using SIMCA–P to identify the differential components between tea samples. All drawings were undertaken using GraphPad Prism 9 and Microsoft Excel 2013.

## 3. Results and Discussion

### 3.1. Sensory Quality

The sensory qualities of the three early spring green teas were investigated. The sensory evaluation findings (Appendix A) showed that the appearance was yellowish-green for all of the samples. The leaf fragments were curly, fairly tippy, and jade green in the HJC and FD samples, and straighter and a more delicate yellow in the AJ ones (Figure 1). A yellow-green and bright liquor color was observed in all three teas but was most brilliant in the HJC. The fresh and tender aroma was present in both HJC and AJ, but a more obvious and persistent chestnut-like odor was present in HJC. A flowery aroma was observed in the HJC and FD samples, and a clean refreshing aroma was noticed in the FD sample, which was described as lasting, gentle, and heavy. In terms of sensory taste quality, all of the teas were described as fresh and brisk, but a mellow taste was most pronounced in HJC and AJ compared to FD. AJ had a fresh and mellow taste, while HJC was fresh and mellow, even heavy and brisk. Infused leaves from all of the samples showed characteristics of brightness and evenness, but AJ had a yellowish color and was softer. The sensory characteristics noted for HJC were as follows: the dry tea was jade green in color; the rope was tight and curly; the liquor was yellowish-green and bright; the aroma was fresh and tender with a high and long chestnut aroma; the taste was fresh, mellow, heavy and brisk; the infused leaves were delicate-green, fat, and bright; and it could be distinguished from the other two kinds of early spring green tea by a traditional sensory evaluation.

### 3.2. Differences in Liquor Color

The relative color values (ΔL*, Δa*, and Δb*) of the three green tea infusions can be seen in Figure 2. There were no significant differences observed between them, indicating that the color quality of early spring green tea infusions was excellent and relatively uniform. PCA (Figure 2f), and PLS–DA (Figure 2f) analyses were conducted on the L*, a*, and b* values, and indicated that the color measurement values could not completely distinguish the three infusions, particularly between HJC and FD.

### 3.3. Identification of Key Aroma Components

#### 3.3.1. Qualitative Analysis of Volatile Organic Compounds

The GC–IMS method was used to determine the VOCs present in the samples. According to the GC–IMS spectral diagram (Appendix A), significant differences in VOC peak signal distributions were observed. Appendix A details the peak intensities of all VOCs in the samples. A total of 84 peaks were detected, 76 of which—including some volatile monomers, dimers, and trimers—were identified by comparing their ion drift and GC retention times against the IMS database. They contained 21 aldehydes (6 dimers), 18 esters (4 dimers), 12 alcohols (2 dimers), 10 alkenes (4 dimers and 1 trimer), six ketones (2 dimers), and nine other categories. In addition, 66, 69, and 65 VOCs were identified in the HJC, AJ, and FD samples, respectively. The total relative VOC contents of the three teas were HJC > AJ > FD.

As is shown in Figure 3, aldehydes had the highest contents in all of the green tea samples, at a range of 35.35–45.90%, followed by esters (15.94–25.96%), and alcohols (15.81–18.47%). Conversely, alkenes and other VOCs were present in extremely low concentrations in all of the samples. In the HJC samples, the esters and alcohols were prevalent in higher percentages than in the AJ and FD samples, yet aldehydes were lower. With regard to the remaining VOC categories, the relative contents in the HJC samples fell between those of the AJ and the FD ones.

#### 3.3.2. Fingerprint Analysis by GC–IMS

To observe the differences in each compound between the different samples, gallery plot analysis was used as a fingerprinting technique (Figure 4). All of the spots in a certain row formed the fingerprint of a sample, while the spots in a certain column display the relative content of a specific compound in different green tea samples. Brighter spot colors thus indicated higher relative content. According to the distributions of VOCs in the samples, each fingerprint was divided into six areas: A, B, C, D, E, and F. Region A was the region shared by the three green tea samples, comprising VOCs that were similar and found in higher abundances. These characteristic VOCs had the typical green tea flavors of grass, fruit, and roasted caramel. Region B showed that the contents of compounds were higher in the HJC samples relative to the AJ and FD ones. It is presumed that this region is composed of the main VOCs that constitute the unique tender chestnut aroma of HJC, such as (E)-2-octenal, linalool, ethyl acrylate, ethyl acetate, and 2-methyl-3-furanethiol. By contrast, the E region is the area where the HJC samples had much lower VOC contents. Region C showed a significantly higher content of compounds in both HJC and AJ. Region D and F are the regions where the AJ and FD samples had significantly higher VOC levels, respectively. The VOCs in these regions were the characteristic components constituting their unique fragrance. The separation of the VOCs from the three green teas can be seen visually in Figure 4, which confirms that the aromas of the three teas differed greatly—a finding that was consistent with our sensory evaluation.

### 3.4. E-tongue Analysis

E-tongue analysis was used to assess the differences in taste between HJC and the other teas. A radar plot (Figure 5a) showed that samples from the three samples presented certain similarities in taste. They were found to differ greatly in terms of astringency, bitterness, and saltiness, but to be similar with regard to other taste indicators. In terms of astringency and bitterness, FD was significantly higher than HJC and AJ, while AJ samples were more prominent in saltiness. HJC therefore fell in the middle of the three. A box plot (Figure 5b) was drawn after removing any indicators that were present below the limit of taste detection. This indicated that the sourness of all of the samples was below the detection limit, while the rest of the tastes were detectable. As can be seen in Figure 5b, the greatest variability between the samples was for the salty taste index, with an intensity extreme difference value of 12.98, followed by astringency, bitterness, and sweetness, with extreme differences of 12.29, 5.87, and 5.30, respectively. The least variability was found for richness. A correlation analysis was conducted on effective taste indicators (Figure 5c), where bitterness, astringency, and umami were correlated with their corresponding aftertaste indicators, while sweetness was not significantly correlated with the other seven indicators. Bitterness was found to be highly correlated with astringency (r = 0.925), and negatively correlated with saltiness and umami (r = −0.939 and −0.568, respectively). A positive correlation was found between umami and saltiness (r = 0.574), and a negative one with astringency and its aftertaste (r = −0.557, −0.667, respectively). These results suggest that there may be some overlapping expression of taste information among multiple taste indicators. Any significant correlations between bitterness, astringency, umami, saltiness, and richness were visualized on bubble charts. As can be seen in Figure 5d, the bitterness and astringency of HJC are much lower than that of FD, but its level of umami is higher. Compared to AJ, HJC has a similar umami value, but its bitterness and astringency are slightly higher. As can be seen from Figure 5e, the saltiness value of HJC is between that of AJ and FD, but its richness is the highest. The samples are clearly distinguished in the PCA shown in Figure 5f. Since these three kinds of early spring green tea can be distinguished by the E-tongue instrument, their differences in taste are indeed significant, which is also consistent with the sensory review.

### 3.5. Analysis of Main Quality Components

The material basis of quality components in tea determines its characteristics, and they are the precise expressions of its sensory properties. To determine the potential of HJC to make high-quality green tea, its quality-related biochemical components, as well as those of AJ and FD, were analyzed and compared. The main quality components of the three early spring green teas are shown in Figure 6, which indicates significant differences between the HJC sample and the two controls. Figure 6a,b show that the total soluble solids and tea polyphenol contents of the three samples were FD > HJC > AJ. Total soluble solids in tea are positively correlated with the number of taste components, which can reflect the strengths and thicknesses of tea infusions [16]. Tea polyphenols are a critical element of total soluble solids, and the main substances that constitute the bitterness and astringency of green tea. They also have a positive impact on improving the concentration of tea infusions [34]. The content of total soluble solids and tea polyphenols was the highest in the FD, which was indicated by the strong and bitter tastes of its tea infusions. AJ’s tea polyphenol content was notably lower than the other two samples, suggesting a low bitterness and bland flavor. The total soluble solid content of HJC was slightly lower than that of FD, but its polyphenol content was in between that of the two varieties AJ and FD. Its tea broth was thick and showed a certain degree of stimulation. This was consistent with the results of our sensory evaluation, detailed in Appendix A.

Catechins are the primary components of tea polyphenols, possessing both bitter and astringent taste properties [35]. The taste threshold of ester catechins (EGCG, GCG, and ECG) is lower than that of non-ester catechins (DL-C, EC, GC, EGC), indicating that the bitterness and astringency of ester catechins are more intense than those of non-ester catechins, and thus these represent the key chemical components that affect the bitter taste of tea [36]. Figure 6c,d demonstrate that the total catechins and ester catechins contents of the three green teas analyzed in this study were FD > HJC > AJ. This result was consistent with that of the E-tongue analysis, which indicated that the bitterness and astringency of HJC tea fell in between that of AJ and FD.

The umami flavor of green tea is largely due to amino acids, which can act as a buffer against the convergence of tea polyphenols. The ratio of polyphenols to amino acids is a key indicator of freshness and briskness in green tea [37,38,39]. As is shown in Figure 6e, AJ had the highest free amino acid content, at 6.25%, followed by HJC at 5.19%, and FD at 4.09%. The phenol-to-ammonia ratio (Figure 6f) showed the opposite trend, with AJ (4.7) < BJ (5.7) < FD (8.5). This indicated that HJC retained a certain degree of convergence, while maintaining high umami.

Caffeine is another important substance responsible for bitterness in tea [40]. It has also been shown that caffeine can improve the freshness of a tea infusion and reduce its bitterness, by hindering the binding of catechins to salivary proteins and thus changing their taste-presenting properties [41,42]. Speculation has been that there is no substantial link between the bitterness of teas and the caffeine content [34,43]. As can be seen from Figure 6h, the caffeine content of HJC was higher than that of AJ and slightly lower than that of FD, which fell in the middle. According to Figure 6i, the total soluble sugar content was the lowest in HJC, but it had little effect on the sweetness of the tea because it was below its taste threshold [44].

The total soluble solid, tea polyphenol, free amino acid, caffeine, total catechin, and ester catechin contents of HJC early spring green tea fell somewhere in between those of the FD conventional control tea and the high amino acid AJ control. HJC also maintained a relatively harmonious phenol-to-ammonia ratio. This corroborates our taste evaluation of the HJC infusions, which were described as fresh, mellow, heavy, and brisk.

## 4. Conclusions

Overall, the sensory characteristics and main quality components of HJC early spring green tea are between those of ordinary green tea and high amino acid green tea, generally falling closer to ordinary green tea. It is difficult to separate them in terms of color, but we found significant differences in VOCs, E-tongue values of bitterness and astringency, and the content of major taste quality components. VOCs such as methyl acetate, linalool, ethyl acrylate, 2-methylpentanal, (Z)-β-ocimene, 2-methyl-3-furanethiol, methyl sulfide, (E)-2-octenal, and ethyl acetate were the main differential components responsible for aroma, and likely influenced the tender chestnut aroma of HJC. In addition, free amino acids, tea polyphenols, and ester catechins were found to be the main differential components responsible for taste, and their harmonious phenol-to-ammonia ratio had a positive effect on the fresh, mellow, heavy, and brisk taste of HJC. This is a valuable finding that provides a scientific explanation for the sensory characteristics of HJC. The results of this study provide not only a reference for normalizing, standardizing and digitalizing quality evaluation of the other kinds of green tea, but a theoretical basis for further improving the quality of HJC and the breeding of similar high-quality tea varieties. With the enrichment of HJC categories, our findings will need to be verified and supplemented in the future to encompass the richness and comprehensiveness of HJC samples. In addition, different grades of HJC have different compositions relating to their maturity. Manufacturing process and form (rolled or powder) affect the quality of green tea equally. In the future, a more comprehensive database of green tea will be established, and big data analytics will be used to predict unknown samples, grade, and other quality factors.

## Figures and Tables

**Figure 1 foods-12-03234-f001:**
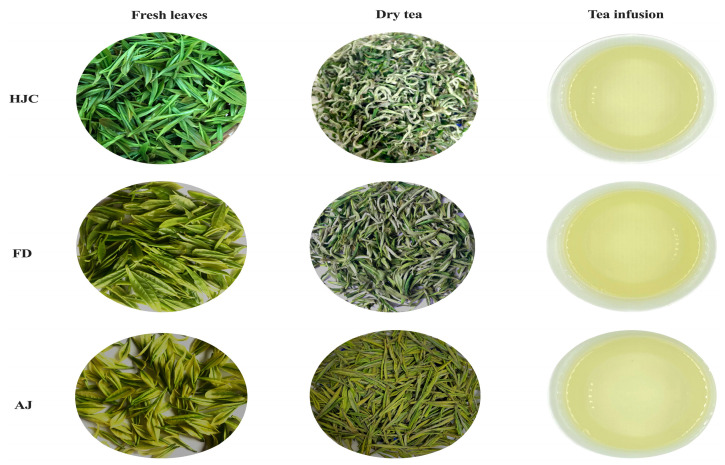
The fresh leaves, dry teas, and tea infusions of the three early spring green teas.

**Figure 2 foods-12-03234-f002:**
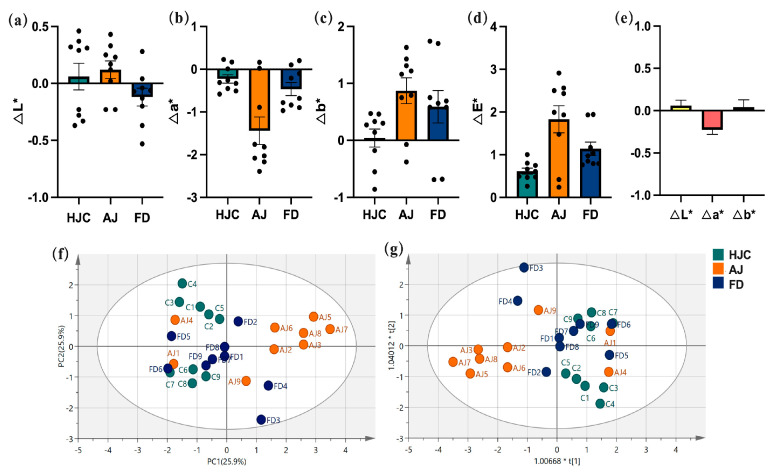
The color values of the three early spring green tea infusions: (**a**) ΔL* value, (**b**) Δa* value, (**c**) Δb* value, (**d**) ΔE* value, (**e**) the relative differences in the color values, (**f**) PCA statistical analysis, and (**g**) PLS–DA statistical analysis. Note: HJC1 was used as a standard sample.

**Figure 3 foods-12-03234-f003:**
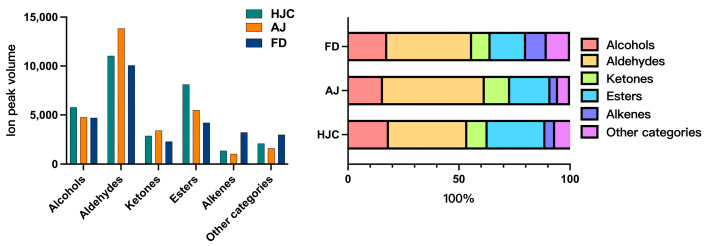
Relative content and proportion of volatile organic compounds (VOCs) in the three early spring green teas.

**Figure 4 foods-12-03234-f004:**
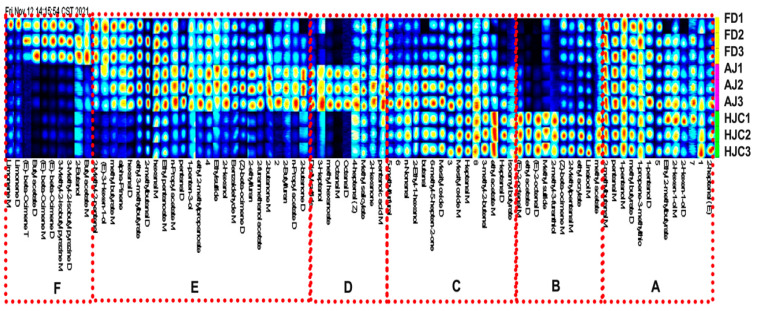
VOC fingerprints of the three early spring green teas.

**Figure 5 foods-12-03234-f005:**
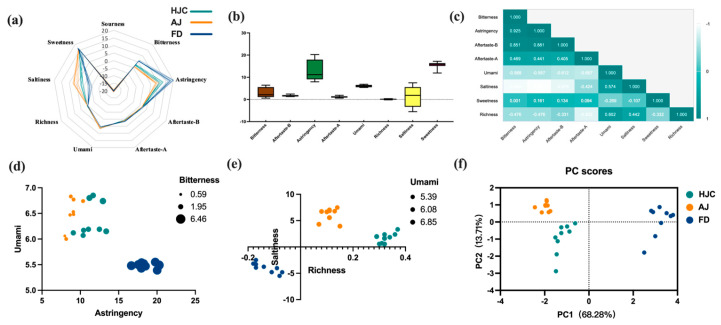
Differences in tastes among HJC, FD, and AJ teas, as indicated by E-tongue analysis: (**a**) radar plot; (**b**) box plots; (**c**) correlation analysis; (**d**) bubble chart of umami, astringency, and bitterness; (**e**) bubble chart of saltiness, richness, and umami; and (**f**) PCA analysis.

**Figure 6 foods-12-03234-f006:**
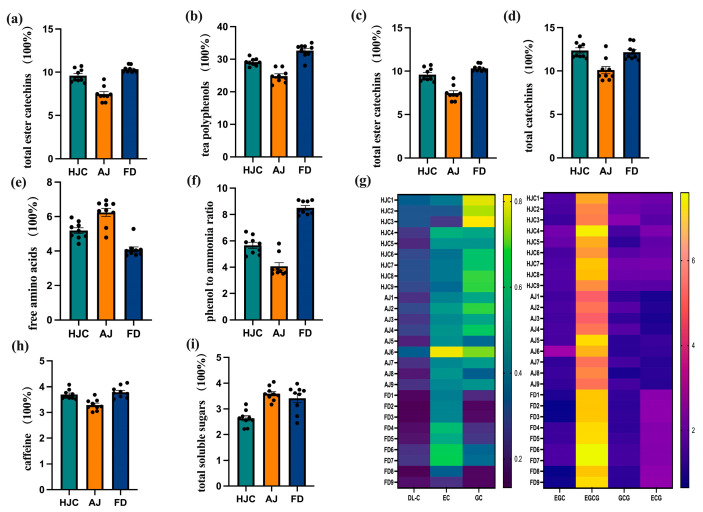
The quality components in HJC, FD, and AJ teas: (**a**) total soluble solids; (**b**) tea polyphenols; (**c**) ester catechins; (**d**) total catechins; (**e**) free amino acids; (**f**) phenol-to-ammonia ratio; (**g**) heat map of catechin components; (**h**) caffeine; and (**i**) total soluble sugars.

## Data Availability

The data used to support the findings of this study can be made available by the corresponding author upon request.

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
