# Peer review of "Characterization of the Sensory Properties and Quality Components of Huangjin Green Tea Based on Molecular Sensory-Omics"

_foods, 2023, doi:10.3390/foods12173234_

Round 1

Reviewer 1 Report

The article entitled “Characterization of the sensory properties and quality components of Huangjin green tea based on molecular sensory-omics”, submitted to the journal presents a study aimed at characterizing the sensory profile of Huangjin green tea and two other Chinese green tea varieties using instrumental methods in relation to conventional, government-standardized sensory assessment. Sensomic profiling of different types of food and beverages is a relatively new approach to assessing the quality and authenticity of foods, helping to gain deeper insight into the link between the profile of sensory active compounds and sensory properties.

The manuscript is clearly structured and written mainly in a language understandable to potential readers. The chosen methodology meets the stated objectives of the study. The results obtained have been satisfactorily presented and discussed.

I have the following remarks on the article:

L 97-106 I recommend a better, clearer specification of the aim(s) of the study at the end of the introduction. The last 2 sentences of the introduction are rather a conclusion.

L189-193 Methods for the determination of total polyphenols, sugars, amino acids should be described.

I would recommend that you consider enlarging the images, in particular Figure 4, as they are very difficult to read as they are.

L374-399 The conclusion is too long, it repeats the results. The conclusion should be relevant to the aim(s) of this study. State better and briefly what new findings have been identified, how these findings increase knowledge in the field, how they are or will be used in practice.

In general, the study yielded a number of results and the obtained findings are useful for further research and future application. This research is in line with current scientific trends.

Author Response

Dear reviewer,

We are thankful to you for your comments and professional advice. These opinions help to improve academic rigor of our article. We have thoughtfully taken into account these comments. The explanation of what we have changed in response to the concerns is given point by point in the following pages.

Point 1:  L 97-106 I recommend a better, clearer specification of the aim(s) of the study at the end of the introduction. The last 2 sentences of the introduction are rather a conclusion.

Response 1: As you suggested, we made a direct statement for the purpose of this study in the first sentence of the last paragraph, and revised those sentences that resemble conclusions.

Point 2: L189-193 Methods for the determination of total polyphenols, sugars, amino acids should be described.

Response 2: The methods for the determination of total polyphenols, sugars, amino acids have been described in many past literatures, and we are afraid that too much description would seem cumbersome. Now these methods have been described in revised manuscript.

Point 3: I would recommend that you consider enlarging the images, in particular Figure 4, as they are very difficult to read as they are.

Response 3: We apologize that we wish to keep the size of the images as we feel them look better this way. The images we provided were vector images which can be arbitrarily scaled and maintain visual quality of them. If you find it a little difficult to read, we recommend enlarging the image.

Point 4: L374-399 The conclusion is too long, it repeats the results. The conclusion should be relevant to the aim(s) of this study. State better and briefly what new findings have been identified, how these findings increase knowledge in the field, how they are or will be used in practice.

Response 4: According to your suggestion, we have streamlined the conclusion and deleted the same content as the results section.

We hope that all these changes fulfil the requirements to make the manuscript acceptable for publication in foods.

Thanks again for your guidance. We believe that the comments have been highly constructive and very useful to restructure the manuscript. 

Sincerely yours,

Ni Zhong

Reviewer 2 Report

The authors investigated the relations between compositional analysis and sensory properties of a special kind of green tea growing in China. They compared three different tea varieties and searched for the effect of polyphenol contents and amino acid content of these tea products on their taste when they were brewed. They used both sensory panel and e-tongue to evaluate and detect the chemical flavor compounds playing role in different taste in brews. They evaluated their results using statistical analysis and possess the significant differences. The study is very interesting to see which compound is relevant to which taste attribute in green tea and results are showing promising contribution to breeding different varieties. My suggestion could be also to take into account the processing effects and form ( rolled or powder) of the tea leaves while evaluating the quality parameters in the further research.

In addition; I have a few remarks below;

Line 236 : Figure 2g should be written for PLAS-DA

Line 398:  Pls add technique name as 'Big data analytics' to 'Big data' in this sentence.

General use of language is very understandable, the manuscripts only requires some corrections in ;

Line 15: Please add 'compare' here '.... HJC and compare to Fuding Dabai' ...

Line 90: The typo in 'Methodology' needs to be corrected.

Line 103-106: especially the verb 'yielding' and 'digitally evaluating' could be rephrased.

Line 203-204: Just as a suggestion, please add short but more descriptive explanation of the methods in the statistical analysis in part 2.8.

Author Response

Dear reviewer,

We are thankful to you for your comments and professional advice. These opinions help to improve academic rigor of our article. We have thoughtfully taken into account these comments. The explanation of what we have changed in response to the concerns is given point by point in the following pages.

Point 1: Line 398:  Pls add technique name as 'Big data analytics' to 'Big data' in      this sentence.                                                                                                                    Line 90: The typo in 'Methodology' needs to be corrected.                                               Line 103-106: especially the verb 'yielding' and 'digitally evaluating' could be rephrased.                                                                                                                         Line 203-204: Just as a suggestion, please add short but more descriptive explanation of the methods in the statistical analysis in part 2.8.                                                          

Response 1: This section has been corrected as your suggestion, and thanks again for your guidance.

Point 2: Line 15: Please add 'compare' here '.... HJC and compare to Fuding Dabai' ...

Response 2: It would be more suitable to change it to your suggestion, but the abstract would be over 200 words. So we are conflicted about whether to change it or not.

Point 3: Figure 2g should be written for PLAS-DA

Response 3: We did use PLS-DA for the analysis of this part. Since the result showed low dispersion, we did not continue to use OPLS-DA to determine the difference in composition between the tea samples.

In addition, the processing effects and form ( rolled or straight) of tea you mentioned are already in our research plan, while we have few studied on powder form. We believe that the suggestions have been highly constructive and very useful for our future research. We hope that all these changes fulfil the requirements to make the manuscript acceptable for publication in foods.

Thanks again for your guidance.

Sincerely yours,

Ni Zhong